# The proximity of ideas: An analysis of patent text using machine learning

**Sijie Feng**⬚*

Department of Economics, NYU Stern School of Business, New York, NY, United States of America

* sfeng@stern.nyu.edu

## Abstract

This paper introduces a measure of the proximity in ideas using unsupervised machine learning. Knowledge transfers are considered a key driving force of innovation and regional economic growth. I explore knowledge relationships by deriving vector space representations of a patent's abstract text using Document Vectors (Doc2Vec), and using cosine similarity to measure their proximity in ideas space. I illustrate the potential uses of this method with an application to geographic localization in knowledge spillovers. For patents in the same technology field, their normalized text similarity is 0.02-0.05 S.D.s higher if they are located within the same city, compared to patents from other cities. This effect is much smaller than when knowledge transfers are measured using normalized patent citations: local patents receive about 0.23-0.30 S.D.s more local citations than compared to non-local control patents. These findings suggest that the effect of geography on knowledge transfers may be much smaller than the previous literature using citations suggests.

**Data Availability Statement:** All replication files are available from Kaggle. doi: 10.34740/KAGGLE/DSV/1157214. Available from: https://www.kaggle.com/fsfeng/replication-data-for-the-proximity-of-ideas.

## Introduction

This paper introduces a measure of proximity in ideas using unsupervised machine learning. I explore knowledge relationships in innovative ideas by: first, deriving vector space representations of patent abstract text using Document Vectors (Doc2Vec); second, using cosine similarity to measure their proximity in ideas space. I illustrate the potential uses of this method with an application to localization in knowledge spillovers.

One explanation for why innovation is concentrated in cities is that knowledge spillovers are geographically constrained. This means that local inventors and firms benefit more from knowledge transfers from other local firms, compared to inventors and firms located in different cities. A prominent literature of measuring knowledge spillovers has emerged from [1] (henceforth JTH) that uses patent citations to study the "paper trail left by the diffusion of innovative knowledge. The general consensus of this literature is that there are large and significant geographic localization effects for knowledge spillovers.

I apply similarity in two different ways. First, I use the standard citations methodology of measuring localization by examining the percentage of local forward citations made to local patents compared to a non-local control, across four decades of observations 1976-2015. Here, instead of selecting the control based on USPC (United States Patent Classification) primary

**Funding:** The author received no specific funding for this work.

**Competing interests:** The author has declared that no competing interests exist.

class, I select a control based on text similarity, which should provide a better proxy of underlying technological proximity between the two patents. While I do indeed find smaller localization effects with the similarity selected control, I still find that local patents receive 0.20-0.27 S.D.s more local citations than that of the non-local control, after normalization. This amounts to approximately a 10-13% reduction in JTH localization estimates. Prior literature [2, 3] has suggested that patent attorneys play a large role in determining patent citations. l control for lawyer effects by selecting a control from the same primary class <u>and</u> lawyer. I find that this reduces localization estimates substantially: local patents receive 0.8-0.12 S.D.s more local citations compared to the non-local control from the same lawyer. This reduces the JTH localization by approximately 60%. This provides a partial explanation for why better measures of technological proximity do not yield lower estimates of localization: citations may be localized in part because lawyers' knowledge of "citable" patents are geographically concentrated. Thus, the majority of localization effects using citations can be accounted for by selecting different controls.

If citations may overstate localization of knowledge spillovers, a different approach may be useful. If local firms and inventors learn from each other's inventions, then within a technology field, patents the same city should express ideas more proximate to each other on average compared to patents from different cities. Further, I find that patents that cite each other have much higher similarity on average, which suggests that a higher incidence of direct knowledge flows does imply greater proximity of ideas. Under this second approach, I find that patents within the same city are 0.02-0.05 S.D.s more similar to each other than patents from other cities, after normalizing text similarity measures. These findings provide further evidence that localization effects may be less than previously thought.

I address the concern that text similarity may be a noisy measure of idea proximity by validating its ability to find large effects on proximity: patents that are from the same primary class, or share a common backward citation, or share an inventor are found to have significantly higher text similarity. Thus, text similarity is not just noise: a noisy measure would find weak estimates across all dimensions. Another concern is that lawyers and examiners may also exert influence on patent text. I find that patents from the same lawyer and processed by the same examiner do have higher text similarity, but that this effect is attenuated when further controls for technology proximity across patent classes are included.

Citations and idea proximity provide different windows into knowledge relationships. Two potential explanations are discussed that may bridge the difference in localization of citations and localization of idea proximity. First, the number of local inventors that influence each other may be very small, which supports the microgeography literature [4, 5] that suggests disconnected clusters of innovation coexist even within the same city. Second, patent text will also reflect to influence of knowledge sources besides other patents, from scientific and other academic publications, to non-codified "tacit" knowledge. While [6, 7] discuss the importance of non-patent knowledge for innovation, patent text does indeed capture the influence of a broader range of potential knowledge flows, which may not be relevant for all studies. These discussions may provide some guidance for applied researchers seeking to understand best use and limitations of text similarity methods. Besides knowledge spillovers, proximity in ideas can potentially have a broad range of other applications. I discuss potential avenues for future research in the conclusion.

This paper provides a contribution to the literature on measures of knowledge and innovation through patent data. Alongside [1], the prior literature using has found significant geographic localization in a variety of contexts: [8] and [9] using spatial distance measures; [10], [11], and [12] using geographic mobility of inventors; [13] with university patents and scientific publications. Only [14] find that localization estimates are insignificant between extremely

technologically proximate patents. Limitations of patent citations are well documented in the literature [15], [16]. Three main critiques are raised: (i) the addition of citations from patent examiners as discussed in [17]; (ii) strategic considerations to add irrelevant citations to block potential infringement suits and to omit relevant citations to broaden the patent scope [18], [19]; (iii) the influence of lawyers on applicant's citations [2], [3]. The evidence I find that localization may be less geographically constrained also contributes to the literature on agglomeration and urban economics, notably in support of [20] and [21] who argued against spatial limitations to knowledge spillovers.

Recently, a small literature has bourgeoned around applying text analysis to patent text. [22], [23] and [24] use a variation of term-document frequency to construct vector representations of patents, which uses the proportional counts of different terms within a patent. [24] adapts their measure to specifically account for the innovativeness of a patent, by overweighting infrequent terms up to the year of appearance of the patent. They also argue for the advantages of using patent text similarity against citations based measures, specifically that citations "given an incomplete representation of which predecessor technologies are important for a new patent." Additionally, [22] validates the accuracy of text-based similarity measures with technology experts.

My contribution differs in that while these other measures use text-based frequency, they do not use machine learning methods that were devised to address shortfalls in frequency vector representations of documents. Primarily, frequency measures fail to account for terms with similar meanings (synonyms such as software and program) and terms with multiple meanings (polysemy such as program). Additionally, these measures do not utilize crucial information in semantic patterns such as the co-occurences in terms across documents. Thus, frequency based approaches may fail to capture the presence of similar ideas expressed through differing semantics and terminology. Finally, frequency measures results in extremely sparse and high-dimensional vectors, which may be computationally expensive. Unsupervised machine learning methods, including Latent Dirichlet Allocation [25], Word2Vec [26, 27], and Doc2Vec [28] were devised precisely to address such concerns. [29] specifically compares frequency measures against each of the unsupervised machine learning methods in how well they each calculated document similarities compared to human annotators. Doc2Vec proved to be the most successful in their evaluation. However, the trade-off to using Doc2Vec is that its selection of text vectors is more black-box and less interpretable, as it utilizes a neural network structure.

Text analysis and unsupervised machine learning has also been applied to patents in non-similarity based contexts. Similar to [24], [30] fit a Latent Dirichlet Allocation model on patent text to determine breakthrough innovation. They find that a topic-originating or breakthrough patent receives approximately 1.4 times more citations than the average patent. [31] also use relevant keywords found in patent abstracts to construct a semantic network classification system to map the technological taxonomy of patents. [32] use a machine learning approach to identify in-text citations within patent text, which they propose as a better measure of direct knowledge flows. [33] use the appearance of new terms to examine how adoption of new technology varies by city size.

## Data and methods

### Data sources

Patent data is taken from PatentsView on all utility patents granted 1976-2016, containing data both on inventors (including unique identifiers and location) and patents (assignee, application date, grant date, primary class and subclass). Bibliographic text data is taken from the

USPTO Bulk Data Products, which has all patent bibliographic text from 1976 to end of 2015. Patent abstracts are taken to be representative of the knowledge contained in patents, as they are a summary of the invention. Citations, lawyer, and examiners data for each patent are also taken from PatentsView. Following prior literature, the patent's location is determined as the MSA where the highest proportion of inventors are located.

**Patent technology fields.** Each patent is assigned three technological fields, with each field being nested in the previous. At the broadest level, an NAICS-based industry classification is given using the USPC to NAICS concordance crosswalk, which delegates each patent to a NAICS category according to its USPTO 3-digit primary classification. Additionally, many patents are also assigned a primary sub class. Patents may also include other discretionary classifications, which I chose to exclude in this analysis. Primary subclasses are nested in primary classes, which are in turn nested in a NAICS industry label. There are over 150,000 subclass labels; 450 class labels, and 33 NAICS industry labels.

## Patent abstracts to vector space representations: Document vectors from Doc2Vec

Using patent abstract texts, I use procedures adopted from the NLP literature to clean and convert text to vector representations (see Text-cleaning for details). The Doc2Vec algorithm was introduced by [28] as a means to meaningfully summarize text contained within documents. It is a straightforward extension of the Word2Vec model of [26, 27], which was developed to represent words meaningfully in a vector space (provide "word embeddings"). Word2Vec was found to be surprisingly powerful in capturing linguistic regularities and patterns, for example that $vec$(Madrid) − $vec$(Spain)+ $vec$(France) is closer to $vec$(Paris) than any other word vector. The objective of Word2Vec is to situate words that have similar meanings close to one another. Similarly, Doc2Vec has the objective of situating similar documents close to one another by placing document vectors (DocVec) close to each other in vector space. To do this, the algorithm uses the "context" around each term in the document to derive a vector representation that maximizes the probability its the appearance. (See S2 Appendix for more details on the algorithm; S2 Fig illustrates diagramatically the inputs and outputs of the algorithm) I implement the algorithm using the `gensim` package in Python [34].

For example, for the sentence "Provides for unattended file transfers", the central word "unattended" has the context ["Provides", "for", "file", "transfers"]. Different sentences will have different context and center words. Before the algorithm is implemented, common words or stop words such as "for" are removed and each word is stemmed to the root. "Provides" and "transfers" become "provid" and "transfer." The document identifier, in this case the patent number "US7502754," is treated as a context word for ever word in the patent. Thus, the context for "unattended" would become: ["provid", "file", "transfer", "US7502754"]. The goal of the algorithm is to select word vectors that maximise the probability of the center word, given the context words. In terms of document vectors, the algorithm will attempt to situate the patent document vector as close as possible to the words within the patent text.

Every word and document is assigned a vector of dimension $N = 100$. This is a rule-of-thumb in the literature, according to [35]. The vectors are optimized using a neural network which maximises the log probability of the appearance of each central word. The resulting vector places words that arise in similar contexts close to each other, and documents that contain similar words close to each other.

## Measuring knowledge spillovers: Cross patent similarity

Cosine similarity has been used to measure technological proximity in [36] and [37], as well as being standard in the NLP literature [38]. Initially, other measures (such as Hellinger distance) were also used but found to be very highly correlated with cosine similarity. The prior literature used vectorizations of patent classes listed for each patent, which had the issues of being of varying lengths with unassigned weights for each class. The primary advantage of NLP patent vector outputs is that they are jointly determined, and position each patent vector relative to all other patents within the corpus. Thus, cross-patent comparisons using NLP vector outputs are much more internally consistent than using vectorizations of patent class selections.

For two patents, $i$ and $j$, the cosine similarity between them is:

$$sim(i,j) = \frac{PV_i \cdot PV_j}{\| PV_i \| \| PV_j \|} \tag{1}$$

Where $PV_i$ is the patent vector representation of $i$. This is preferred to Euclidean distance as it is factors in the "size" of the vector; a Euclidean distance measure would assign positive distance to two vectors that contained the exact same words, but of different quantities. Cosine similarity normalises all measures to be in the range $[-1, 1]$.

**Number and proportion of common backward citations.** Since the incidence of direct citation between two random patents are rare, a measure of "indirect" knowledge flow between the two patents would be the number or proportion of common backward citations between two patents:

$$ncc(i,j) = |\{citations_i\} \cap \{citations_j\}| \tag{2}$$

$$pcc(i,j) = \frac{|\{citations_i\} \cap \{citations_j\}|}{|\{citations_i\}|} \tag{3}$$

Where $ncc(i,j)$ represents the number of common backward citations between patents $i, j$ and $pcc(i,j)$ the proportion of backward citations of $i$ that were also made by $j$. For example, if patent $i$ cites $\{A, B, C, D\}$, and $j$ cites $\{D, E\}$, $ncc(i,j) = 1$ and $pcc(i,j) = 0.25$. In each case, self-citations are removed first. These variables can be thought of as measuring the degree of similitude in the patent knowledge sources of the two patents using a citations-based approach.

**Technological field proximity.** Since each patent is assigned technology field labels in the form of primary classes, technological field proximity between two primary classes can be measured using the average similarity of a sample of patents in each primary class. For each year $t$, I take a sample of up to 1000 patents in each primary class pair $pc_i, pc_j$ that were granted in the previous 5 years. I then calculate the mean of the pairwise similarities between all such pairs. Thus:

$$sim(pc_i, pc_j)_t = mean\Big( \{sim(i,j) | i \in pc_i, j \in pc_j\}_{t-5,t} \Big) \tag{4}$$

Intuitively, this represents the expected similarity between two patents if only their technology field was known. Cross field similarity are analogous to the technological proximity measures of [37, 39]. Both papers, alongside other citations-based methods of measuring technological proximity, rely on the vectorization of PTO classes. These methods may lead to inconsistent results as each patent may have any number of non-primary classifications. The standard procedure has been to normalize or weight each of the classes listed, which discretizes the vector space and leads to discontinuities in the proximity measures. (A patent with one

**Table 1. Average DocVecs similarity for pairs of patents that match on each column.** The standard deviation of the similarity in that sample reported in the next column. Samples are partitioned by decade.

| Year Group | NAICS Match | S.D. | Primclass Match | S.D. | Inventor Match | S.D. | Direct Citation | S.D. | Year Match | S.D. |
|---|---|---|---|---|---|---|---|---|---|---|
| 1975-85 | 0.126 | 0.137 | 0.187 | 0.145 | 0.301 | 0.148 | 0.328 | 0.148 | 0.126 | 0.138 |
| 1985-95 | 0.124 | 0.135 | 0.186 | 0.145 | 0.320 | 0.163 | 0.322 | 0.146 | 0.124 | 0.135 |
| 1995-05 | 0.129 | 0.134 | 0.196 | 0.147 | 0.312 | 0.158 | 0.302 | 0.147 | 0.129 | 0.134 |
| 2005-15 | 0.141 | 0.136 | 0.200 | 0.146 | 0.310 | 0.170 | 0.300 | 0.152 | 0.141 | 0.136 |

class would be represented by a vector with 1 in the class column and 0 elsewhere; two classes 0.5 in each class column and elsewhere; and so on).

**Validating similarity measures.** Prior expectations about patent similarity can be used to validate the vectors generated by the Doc2Vec algorithm. In (Table 1) the baseline group average is the average pairwise similarity for patent pairs from within the same NAICS industry granted within 5 years of one another. We should expect that, on average, similarity between patent pairs of the same primary class should be <u>higher</u> than pairs within the same NAICS industry, since industry represents a broader definition of technology field. (Table 1) shows that patents within the same primary class have average similarity around 1.5 times that of patents just within the same NAICS industry. Patent pairs sharing an inventor have 2.5 times the similarity of the baseline group. Patent pairs that have a direct citation relationship also have a comparable level of similarity to patent pairs sharing an inventor. On the other hand, we should also expect that patent pairs from the same grant year should not have average similarity higher than the baseline, since the time difference between 0 years and 1-5 years is not large enough to have a significant impact on technological difference. (Table 1) shows there is virtually no difference between average similarity of patents granted in the same year and the baseline. In general, variance is higher in smaller samples such as patent pairs with matching inventors. Since DocVecs captures trends in similarity that matches prior expectations, it is unlikely that results are being driven by noise in the vectors generated by the algorithm.

## Application of similarity: Estimating geographic localization

The similarity measure can be applied in two ways to estimate geographic localization of knowledge spillovers. In the first case, I replicate and extend the work of JTH up to recent years. This standard methodology involves the selection of a control patent that is as close as possible in grant date to the "target" patent, within the same primary class. I then select different control patents using (i) patent text similarity; and (ii) a patent from the same primary class and the same lawyer. While selecting a control based on similarity does not drastically alter the estimates for localization, selecting on lawyer does significantly diminish localization estimates up to 2005.

In the second case, I look for evidence of localized knowledge spillovers by estimating whether within-technology field patents from the same MSA are more similar than if they are from different locations. The rationale is that if firms and inventors from the same "cluster" (defined as a technology field within an MSA, for example Pharmaceuticals in Philadelphia) are learning more from knowledge generated by each other, then the similarity of patents <u>within</u> a cluster should be higher than similarity of patents <u>across</u> clusters (for example, similarity of patents from Pharmaceuticals in Philadelphia to Pharmaceuticals in Boston). I find much less evidence of localization when examining patent text similarity.

## First application: Selecting different controls under standard citations methodology

I replicate and extend the work of JTH in order to have a baseline estimate of localization effects. JTH sampled patents in their control (target) group in the following manner: from the years 1975 to 1980, they select a random sample of Top Corporate (top 200 by R&D total expenditure measured by Compustat) and Other Corporate patents, and all patents granted to Universities. Their sample size is 950 for 1975 and 1450 for 1980 respectively. Then, for each "target" patent in the sample find a control patent that is as close as possible to the target in grant date in the same patent primary class. JTH claim that this accounts for the "existing distribution of technological activity," and thus if forward citations are more likely to be from the same geographical area as the target patent over the control, then it is evidence for the existence of localized knowledge spillovers.

I replicate this method using a larger sample of target patents granted 1976-2005 and limit forward citations to be within 10 years of the target patent's grant date. 2005 is the last year that 10 year forward citations are available for. Self-citations of patents granted to the same assignee are similarly excluded. The only point of departure is that due to lack of data, I do not use separate categories of patents by assignee "type", and pool all patents by grant year. Compared to the original JTH results (table III, p. 590), my results are fairly well aligned with their 1980 cohort figures for top corporate patents: 8.8% for target match and 3.6% for control match; compared to 9.09% and 3.77% for my results. Slight discrepancies may arise due to sample selection and slight differences in removing self-citations.

The next step is to select a different control patent. In the first substitution, I determine a control that is the patent with the highest similarity to the target from a different MSA and a proximate grant date. If we interpret patent text similarity as a better reflection of unobserved technological proximity, then this method may provide better control than merely selecting on PTO primary class, as challenged previously by [14]. This approach is in line with previous attempts such as [22], who also use a text-based similarity measure to select better control patents.

The second substitution follows a separate line of concern. I select a control patent in the same primary class, different primary class, and different assignee to the target (same as in the JTH match), but also from the same lawyer as the target. Previous literature such as [2] has drawn attention to the large effect that patent attorneys play in deciding citations for patent applications. Therefore, localization patterns may be overly influenced by the patent knowledge of lawyers rather than knowledge flows across inventions. If patent lawyers do not have an important role to play in determining the localization of patent citations, then further selecting a control that matches on both primary class and attorney should not yield very different results for localization, compared to the baseline replication. However, if these results prove significant, this may explain why better measures of technological proximity do not yield lower estimates of localization: citations may be localized in part because lawyers' knowledge of "citable" patents are geographically concentrated, not necessarily because knowledge flows across patents are.

Once a control in each case has been selected, I calculate the percentage of forward citations matching the target's MSA for both the target and control. Under this method, localization is significant if the target patent has more local citations compared to the control. Results for the percentage of forward citations matching the target's MSA under each control selection method is presented below in (Table 2).

**Localization under different controls.**   Control selection using text similarity does improve in accounting for unobserved technological proximity, resulting in lower estimates

**Table 2. Baseline results for JTH replication under different control selection methods.** Each column represents the average percentage of forward citations to the target or control in the target's MSA, for patents granted within a certain decade. Sample sizes vary due to the inability to find control patents under certain methods of selection.

| | 1975-85 | 1985-95 | 1995-05 |
|---|---|---|---|
| Control Selection: Standard JTH | | | |
| Target, Pct Cite in Target MSA | 9.1 | 9.7 | 11.0 |
| Control, Pct Cite in Target MSA | 3.8 | 3.5 | 4.5 |
| Ratio | 2.4 | 2.8 | 2.4 |
| p-value | 0 | 0 | 0 |
| N | 58647 | 107358 | 185154 |
| Control Selection: Similarity | | | |
| Target, Pct Cite in Target MSA | 9.2 | 9.7 | 11.0 |
| Control, Pct Cite in Target MSA | 4.8 | 4.1 | 5.1 |
| Ratio | 1.9 | 2.4 | 2.0 |
| p-value | 0 | 0 | 0 |
| N | 36917 | 67332 | 117137 |
| Control Selection: Lawyer | | | |
| Target, Pct Cite in Target MSA | 9.4 | 10.0 | 11.5 |
| Control, Pct Cite in Target MSA | 7.5 | 7.9 | 8.9 |
| Ratio | 1.3 | 1.3 | 1.3 |
| p-value | 0 | 0 | 0 |
| N | 22914 | 51837 | 85855 |

for localization. However, these effects are still highly significant, with local citations being 0.9-1.4 times higher for the local target patent compared to the non-local control. Interestingly, once we select a control that is from the same lawyer as the target, localization estimates shrink dramatically. Local citations are now only 0.3 times higher for the local target. These findings confirm the important role of lawyers in determining how localized citations are, which "muddies the waters" of determining the size of local knowledge flows. Hypothetically, there are a number of mechanisms by which lawyers could bias citations towards localization: (i) lawyers operating in a select few cities may cite the patents of their clients, who are likely to be operating in similar technology fields to begin with; (ii) lawyers may cite the patents of other firms and inventors within these cities that they have encountered either through their own networks or other transactions.

The confounding factor is that many lawyers only operate within one city, representing a handful of firms. Thus, it may be difficult to disentangle the localization of the inventor's knowledge flows (which citations should proxy), from the localization of the lawyer's knowledge of patents (which add noise and bias to the citations measure). Since I find that lawyers representing technologically similar firms across different cities cite significantly more patents in the target's city, this suggests that the size of the bias from lawyers towards localization of citations is not small.

**Regression model for estimating localization.** The above exercise can be represented as a regression model in the form:

$$pct\ cites\ in\ MSA_{T,i} = \beta_0 + \beta_1 I(MSA_i = MSA_T) + X_i + \epsilon \tag{5}$$

Where $i \in \{T = target, C = control\}$. Here, if patent $i$ is the target patent, the indicator $I(MSA_T = MSA_T) = 1$, while for the control patents $I(MSA_C = MSA_T) = 0$.

To account for the potential effect of other variables on citations and localization, $X_i$ represents further controls for the patent $i$, including year, primary class, MSA, lawyer, and examiner fixed effects. Only the 100 largest in each category are included to reduce dimensionality in the covariates matrix. The percentage of citations is normalized prior to the regression, so that $\beta_1$ represents the increase in the (standardized) percentage of local citations when the patent is also local. For consistency, I normalize all three samples under each control selection regimes using the standard JTH control sample in S2 and S3 Tables.

Estimates of localization for fixed effects including year and primary class are presented in S2 Table; for all fixed effects, results are presented in S3 Table. The size of the localization estimates are not particularly sensitive to the inclusion of further controls. In the standard JTH control selection, local patents are found to receive 0.23-0.30 S.D.s more local citations compared to the non-local control. This is diminished by 10-13% to 0.20-0.27 S.D.s in the sample where similarity is used to select the control. Finally, selecting on the same lawyer reduces localization estimates further by 60% to 0.08-0.12 S.D.s. Thus, the majority of the localization effects under JTH can be accounted for through better control selection.

## Second application: Evidence of localization in the proximity of ideas

The findings from the previous section suggest that citations may overstate localization due to the influence of lawyers. But the bias towards localization of citations does not necessarily imply that knowledge spillovers themselves are not localized—it suggests that it may be useful to address the question of localization from a different approach. If knowledge spillovers localized, then this suggests that local firms and inventors learn from each other's inventions. Thus, patents from a particular cluster should express ideas more proximate to each other compared to patents from differing clusters. In (Table 1), patent pairs that have a direct citation relationship are found to have much higher similarity on average, which suggests that direct knowledge flows imply greater proximity of ideas.

Patent text similarity may also be particularly suited to picking up knowledge that were "in the air", tacit knowledge, or common knowledge inputs other than citations. While patent citations reflect knowledge flows from other patents, patent text should reflect the influence of patent, non-patent, and tacit knowledge flows. [7] find a significant role for geographic distance acting as a barrier for such tacit knowledge. In their study of patent interferences, the simultaneous instances of identical invention by two or more independent parties, they find that interfering patents are much more likely to arise from the same geographic location. If, as proposed by [7] and [40], tacit knowledge flows are geographically bounded, then we should find even more proximate ideas within a cluster. For further discussion of the relationship between knowledge flows and idea proximity, see the "Discussion of patent text similarity" section.

**Sample construction.** For patents within the same technology field (either a NAICS industry or a PTO primary class), I sample patent <u>pairs</u> within the same MSA (i.e. patents from the same cluster), and patent pairs from different MSAs (across clusters). Patents from the same MSA are slightly over sampled to ensure a sizable number of patent pairs from the same MSA across a range of technological fields. Patent pairs are granted within 5 years of each other and are assigned to different firms. (N.B.: While some patent pairs may have the same target patent, the number of appearances made by multiples of the same patent is extremely small relative to the entire sample, thus curtailing the presence autocorrelation). Heteroskedastic-robust standard errors are used in regression estimates.

The use of both industry level and primary class level technology fields allows me to capture potential differences in the dynamics of knowledge spillovers. If knowledge spillovers are more localized for firms within the same industry, then patent text within the same industry cluster

**Table 3. Average similarity of patent text within and across clusters.** Within cluster implies patent pairs in the sample are from the same MSA, as well as the same technology field. Across clusters implies patent pairs are from different MSAs. All patent pairs are granted within 5 years of each other, and are assigned to different firms.

| Year Group | 1975-85 | 1985-95 | 1995-05 | 2005-15 |
|---|---|---|---|---|
| Technology field: NAICS Industry | | | | |
| Within Cluster, $I(MSA\ Match) = 1$ | 0.127 | 0.127 | 0.133 | 0.145 |
| Across Clusters, $I(MSA\ Match) = 0$ | 0.124 | 0.12 | 0.125 | 0.137 |
| Ratio | 1.02 | 1.059 | 1.062 | 1.056 |
| $p$-value | 0.001 | 0 | 0 | 0 |
| $N$ | 194131 | 282112 | 443885 | 578056 |
| Technology field: Primary Class | | | | |
| Within Cluster, $I(MSA\ Match) = 1$ | 0.197 | 0.193 | 0.195 | 0.197 |
| Across Clusters, $I(MSA\ Match) = 0$ | 0.19 | 0.182 | 0.184 | 0.188 |
| Ratio | 1.036 | 1.059 | 1.063 | 1.044 |
| $p$-value | 0 | 0 | 0 | 0 |
| $N$ | 171893 | 252886 | 407176 | 537878 |

(i.e industry-MSA pair) should be more similar. Further, if we expect knowledge to be more specialized across clusters at the industry level, then patent text should be more similar within an industry cluster compared to within a primary class cluster.

**Proximity of ideas within cluster and across clusters.** The unconditional sample means for within cluster and across clusters patent text proximity are reported in (Table 3). While the similarity of patents within the same cluster are higher than patents across clusters, the effects are relatively modest: on average, within cluster patent pairs have text proximity 0.02-0.06 higher than patent pairs across clusters. These results are more closely aligned with the conjecture that citations over-estimate the localization of knowledge spillovers.

A number of important limitations should be considered. First, there is the concern that rather than there being limited localization in idea proximity across patent clusters, document vector similarity itself is a poor measure of idea proximity. Mirroring the analysis in (Table 1), I address this concern below. Second, there may be other factors affecting patent text similarity besides knowledge spillovers. For example, both patent lawyers and patent examiners may affect the abstract text. This is further discussed in section 2.2.3. Finally, there is likely to be simultaneity bias between the location of the patent and the similarity of patent text, if firms from similar technology fields are also more likely to collocate. However, this bias is likely to be positive and implies that the effect of local knowledge spillovers should be even smaller than in the reported results. These concerns can be partially addressed through moving to a regression model framework and including suitable control variables.

**Regression model for estimating localization in idea proximity.** In regression form, I estimate for each technology field sample:

$$sim(i,j) = \beta_0 + \beta_1 I(MSA\ Match_{i,j}) + X_{i,j} + \epsilon_{i,j} \tag{6}$$

Here $I(MSA\ Match_{i,j}) = 1$ if patent $i, j$ are from the same location (i.e. $MSA_i = MSA_j$). Similar to Eq (5), $X_i$ represents further controls including year, primary class, MSA, lawyer, and examiner fixed effects. I also include other match controls for the patent pair, which may affect the similarity in their patent text: $I(Lawyer\ Match)$, if patents are assigned to firms that share the same lawyer; $I(Inventor\ Match)$, if patents share an inventor (after inventor relocates to different firm); $I(Primclass\ Match)$, if patents are from the same primary class (only for patent pairs within the same NAICS industry). Fixed effects are for patent $i$ only, to reduce

dimensionality. Match effects depend on both patents. Similarity is also normalized prior to regression. The estimated localization in idea proximity is given by $\beta_1$ and represents how many S.D.s more similar patents are from within a cluster compared to across clusters.

**Validation: Is similarity a noisy estimate for idea proximity?**   One concern may be that instead of geographic proximity being a weak determinant of patent proximity, in fact similarity is not a good measure of patent proximity due to noise. I address this concern by seeing if other match variables are strong determinants of similarity across patents. If similarity is a poor indicator of patent proximity, then we should expect matching on these dimensions to also produce small and possibly imprecise estimates of their effects. Using prior expectations, I expect the following match variables to have a significant effect on patent similarity for patents within the same NAICS industry: $I(InventorMatch)$, if patents share an inventor (after inventor relocates to different firm); $I(PrimclassMatch)$, if patents are from the same primary class; and $I(CommonCited \geq 1)$ to indicate the presence of at least one common cited patent between the pair.

Results for the estimated effect of matching on other variables are reported in (Table 4). I find that estimates for these other match effects are large and significant. The estimated effect of sharing an inventor increases similarity by 1.27-1.52 S.D.s Sharing a common cited patent increases text similarity by 0.84-1.51 S.D.s. Note that the rise in citation rates in recent decades has meant that sharing a common cited patent has declined in effect on DocVecs similarity. Patents from the same primary class have 0.40-0.44 S.D.s higher text similarity. Thus, it is reasonable conclude that similarity is not a noisy estimate for idea proximity, as it is able to pick up on the proximity of patent text across differing dimensions. The effect of matching on the same location cluster may indeed be small.

**Validation: How do lawyers and examiners affect patent text?**   Another related concern is that patent text is also subject to the external influence of lawyers and examiners. Related to the exercise in the above section, I check for their effect by seeing how much text similarity increases when two patents have the same attorney or were processed by the same examiner. However, lawyers and examiners are not assigned randomly: they are both either selected or assigned in a manner correlated with the technology field of the patent. Therefore, omitting the effect of technological proximity across patents may overstate the influence of lawyers and examiners on patent text. I control for technological proximity using Eq (4), that is, by

**Table 4. Effect of matching on other variables for patents within the same NAICS industry.** Estimates are the increase in S.D.s of similarity when matching on each variable is true. Standard errors are reported below in parentheses.

|  | 1975-85 | 1985-95 | 1995-05 | 2005-15 |
| --- | --- | --- | --- | --- |
| $I(InvMatch)$ | 1.2789*** | 1.5206*** | 1.3817*** | 1.2664*** |
|  | (0.1042) | (0.0713) | (0.0559) | (0.0563) |
| $N$ | 192841 | 281222 | 437685 | 569252 |
| Adjusted $R^2$ | 0.07 | 0.07 | 0.08 | 0.06 |
| $I(CommonCited \geq 1)$ | 1.5084*** | 1.2870*** | 1.1149*** | 0.8388*** |
|  | (0.0963) | (0.0637) | (0.0398) | (0.0254) |
| $N$ | 192841 | 281222 | 437685 | 569252 |
| Adjusted $R^2$ | 0.07 | 0.07 | 0.08 | 0.06 |
| $I(PrimclassMatch)$ | 0.4413*** | 0.4449*** | 0.4291*** | 0.4045*** |
|  | (0.0085) | (0.0066) | (0.0050) | (0.0042) |
| $N$ | 192841 | 281222 | 437685 | 569252 |
| Adjusted $R^2$ | 0.08 | 0.09 | 0.10 | 0.07 |
| Controls: Year and PC FEs | | | | |

**Table 5. Regression estimates for the localization of idea proximity, which represents how many S.D.s more similar patents within cluster (location-technology field) are compared to across clusters.** Standard errors of estimates are reported in parenthesese below. A separate sample is computed for each definition of technology field at the industry and primary class level.

| | 1975-85 | 1985-95 | 1995-05 | 2005-15 |
|---|---|---|---|---|
| Technology Field: NAICS | 0.0171*** | 0.0354*** | 0.0344*** | 0.0333*** |
| | (0.0053) | (0.0043) | (0.0034) | (0.0030) |
| $N$ | 192773 | 280962 | 437405 | 563881 |
| Adjusted $R^2$ | 0.08 | 0.09 | 0.10 | 0.06 |
| Technology Field: Primary Class | 0.0277*** | 0.0502*** | 0.0531*** | 0.0395*** |
| | (0.0066) | (0.0052) | (0.0039) | (0.0033) |
| $N$ | 170564 | 251218 | 400729 | 518334 |
| Adjusted $R^2$ | 0.07 | 0.08 | 0.08 | 0.06 |
| Controls: Year, PC, MSA, Examiner, Lawyer Match and FEs | | | | |

calculating the mean similarity of prior patents in each patent's respective primary class. Note that in the regression results presented in (Table 5), matching on lawyer and examiner has already been included as a control. These estimates are to ascertain how large of a role examiners and lawyers have to play in shaping patent text.

Results are presented in (Table 6). Prior to controlling for technology proximity and all other fixed effects, patent pairs from the same lawyers and examiners do indeed have substantially higher text similarity. However, after controlling for the proximity in the patents' respective technology fields, the effect of lawyers and examiners on patent text similarity does decrease substantially. Lawyers, echoing results in the first application section, have a much larger effect on similarity: patents from the same lawyer have 0.14-0.24 S.D.s more similar text than patents from different lawyers, even after controlling for all other effects. The text of

**Table 6. Effect of matching on lawyer and examiner for patents within the same NAICS industry.** Estimates are the increase in S.D.s of similarity when matching on each variable is true. Standard errors are reported below in parentheses.

| | 1975-85 | 1985-95 | 1995-05 | 2005-15 |
|---|---|---|---|---|
| Controls: Year and PC FEs | | | | |
| $I(LawyerMatch)$ | 0.2720*** | 0.3721*** | 0.4429*** | 0.3694*** |
| S.E. | (0.0366) | (0.0281) | (0.0299) | (0.0276) |
| $N$ | 192773 | 280962 | 437405 | 563881 |
| Adjusted $R^2$ | 0.06 | 0.07 | 0.08 | 0.05 |
| $I(ExaminerMatch)$ | 0.4571*** | 0.4867*** | 0.4170*** | 0.4158*** |
| S.E. | (0.0184) | (0.0165) | (0.0186) | (0.0213) |
| $N$ | 192773 | 280962 | 437405 | 563881 |
| Adjusted $R^2$ | 0.07 | 0.07 | 0.08 | 0.05 |
| Controls: Tech proximity and all other FEs | | | | |
| $I(LawyerMatch)$ | 0.1884*** | 0.1460*** | 0.2362*** | 0.1811*** |
| S.E. | (0.0468) | (0.0256) | (0.0271) | (0.0260) |
| $N$ | 102330 | 280954 | 437386 | 563865 |
| Adjusted $R^2$ | 0.11 | 0.12 | 0.12 | 0.08 |
| $I(ExaminerMatch)$ | 0.1334*** | 0.1555*** | 0.0897*** | 0.0769*** |
| S.E. | (0.0249) | (0.0161) | (0.0181) | (0.0205) |
| $N$ | 102330 | 280954 | 437386 | 563865 |
| Adjusted $R^2$ | 0.11 | 0.12 | 0.12 | 0.08 |

**Table 7. Estimates of localization in ideas proximity including technology proximity and all other controls for patent pairs in the same NAICS industry.** Due to a lack of patent data prior to 1976, technology proximity is only available 1980 onwards. Eq 1 uses primary class similarity as a separate control; (2) includes interaction effects with the location match indicator.

| | (1) | | | | (2) | | | |
|---|---|---|---|---|---|---|---|---|
| | **1980-85** | **1985-95** | **1995-05** | **2005-15** | **1980-85** | **1985-95** | **1995-05** | **2005-15** |
| $I(MSA\ Match)$ | 0.0170 | 0.0390*** | 0.0300*** | 0.0274*** | 0.0178 | 0.0415*** | 0.0277*** | 0.0222*** |
| | (0.0120) | (0.0070) | (0.0048) | (0.0038) | (0.0120) | (0.0069) | (0.0048) | (0.0043) |
| $I_{MSA}{}^{*}sim_{DV}(pc_i, pc_j)$ | | | | | -0.0012 | -0.0045 | 0.0050 | 0.0085* |
| | | | | | (0.0116) | (0.0067) | (0.0047) | (0.0045) |
| $sim_{DV}(pc_i, pc_j)$ | 0.2805*** | 0.2913*** | 0.2816*** | 0.2932*** | 0.2808*** | 0.2925*** | 0.2804*** | 0.2908*** |
| | (0.0090) | (0.0055) | (0.0040) | (0.0036) | (0.0095) | (0.0057) | (0.0042) | (0.0039) |
| N | 40323 | 110982 | 215861 | 344313 | 40323 | 110982 | 215861 | 344313 |
| Adjusted $R^2$ | 0.12 | 0.13 | 0.13 | 0.08 | 0.12 | 0.13 | 0.13 | 0.08 |
| Controls: technology proximity and all other controls | | | | | | | | |

patents processed by the same examiner have 0.08-0.16 S.D.s higher similarity; it appears that their effect has declined in the last three decades.

These results indicate that including technology proximity in primary class would attenuate some biases towards higher similarity in patent text due to the influence of external parties.

**Including technology proximity in estimating localization of idea proximity.** For patents within the same NAICS industry, including prior similarity across primary classes may be able to address some concerns about other factors that may raise text similarity, besides knowledge spillovers. It may also partially address the simultaneity bias between text similarity and collocation. In (Table 7), the estimate of localization diminishes further with the inclusion of primary class similarity as a control for technological proximity: the estimate of localization ranges from insignificant to 0.04 S.D.s above the mean; the interaction effect is insignificant at the 5% level across all decades.

## Summary of results

I applied similarity to the examination of localized knowledge spillovers in two ways. In the first application, I replicate JTH's original methodology and use text similarity to find a control patent. While estimates of localization in citations do diminish, suggesting that selecting a control using similarity does a better job of addressing unobserved technological differences in the target and control patent, I also find that localization in citations diminish substantially more when selecting a control from the same lawyer. This complicates the validity of the experiment in identifying the localization in knowledge spillovers, as it indicates that lawyers' knowledge of patents may drive the geographic concentration in citations. As an alternative test, I investigate the localization in idea proximity of patent text: examining whether patents from within a cluster (the same technology field and city) are more similar than patents across clusters. I find that localization estimates are weak, suggesting that citations may in fact overstate localization in knowledge spillovers.

## Discussion of patent text similarity

The results from the previous applications showed that knowledge relationships may appear quite different under examination using the proximity of ideas versus standard citations measures. In light of those findings, I discuss some limitations and implications of these results.

The discussion in this section may also help applied researchers who want to further understand what text and text similarity may reflect.

## Patent abstract vs claims text

The main drawback with using the abstracts of patents may be written to be intentionally general or vague, and may not express the core content of the patent, as examiners focus on assessing claims text when deciding patentability. However, this may also be something of a "reverse advantage" for abstracts, as they are not overly scrutinized and potentially altered by both lawyers and examiners.

To address whether or not the use of abstract text biases the size of localization estimates, I repeat the exercise from (Table 5) using the first five claims in the patent. This serves as a summary of the main contributions of the patent, since claims are listed in consecutive order, with the most important claim first. Results are shown in S4 Table. The geographic localization effect is of a comparable magnitude to using abstracts. Selecting patents within both NAICS industry and primary class yields a localization estimate of 0.04-0.05 S.D.s, indicating that claims text has slightly higher similarity within the same cluster than compared to abstract text. However, this is still far lower than the size of the estimate by JTH.

## Relationship between text similarity and knowledge

**High similarity does not reflect direct knowledge flows.** The results from the previous section relied on the argument that if a group of patents have a high incidence of shared knowledge flows, then the similarity of text within such a group should be high. For example, patent pairs that have a direct citation relationship will have higher similarity on average (Table 4). However, an individual patent pair with high similarity does not imply that a direct knowledge flow has taken place. Inferring the presence of localized knowledge spillovers using text similarity is appropriate as it relies on comparison across aggregate or group means. If accurate indications of direct knowledge flows for individual patent pairs are required for the research question, for example if the focus was on patent and inventor networks, then citations may still be the more appropriate measure.

**Can high local knowledge spillovers lead to weak local idea proximity?** Is it possible that local knowledge flows may still lead to dissimilar local ideas? This may be true in some cases where learning about the innovation agenda of local rivals may lead others to differentiate their inventions. However, it is still the case that on average, a patents that have a direct citation relationship have high similarity, which is to say that patents taking knowledge from other patents should still express more proximate ideas. This does imply that it may not be possible to determine whether low similarity patent pairs have differentiated their inventions, or are simply unrelated.

**Interpreting weak local idea proximity.** One way to interpret the findings of weak local idea proximity is that inventors may be directly influenced by very few other local inventors. This may support the literature on microgeography, which suggests that "what appears to be a cluster at the county level may indeed be several geographically (and often technologically) distinct clusters, each with different social relationships and unique needs." [4] The question is whether or not this implies localization at the city level is high or low. As a simple example, consider inventor $i$ from city $A$ that has 100 inventors all operating in the same technology field. Suppose inventor $i$ exchanges knowledge with 5 other inventors local to city $A$ and 5 inventors from other cities. If we were to examine the proportion of inventor $i$'s influences in city $A$, then localization from this perspective would be large at 0.5. However, if we were to examine the proportion of inventor $i$'s influences relative to all other possible influences in city

*A*, localization would be much smaller at 0.05. Thus, the choice of denominator may decide the magnitude of localization effects.

Another way to look at it would be whether or not "counter-evidence" of knowledge spillovers are important to consider, that is, the lack of knowledge flows where they should exist. Patent citations and other approaches such as interference are more or less silent about this counter-evidence, whereas similarity (which can be generated for any two patents) captures the presence of low proximity in local ideas. Ultimately, it may be left up to researchers to decide whether or not this counter-evidence is important. Even in the case when it is not, similarity can be useful in identifying which knowledge relationships can be disregarded. In terms of findings on the localized knowledge spillovers, even if similarity is considered too "broad" of a measure to accurately capture relevant knowledge flows, the evidence in the JTH replication (the first application) still suggests that localization is likely overestimated using citations.

**Patent texts reflect non-patent knowledge influences.**  Continuing with the discussion from the second application, another reason why localization in idea proximity may be weak is that inventors are highly influenced by other sources knowledge, which may be commonly available and thus non-local (for example, knowledge acquired from the internet). Other sources of knowledge include: academic and scientific publications, textbooks, technical journals, and less concrete examples such as tacit knowledge, background knowledge, and knowledge "in the air". Recent literature by [6, 7] and [33] emphasize the role that external knowledge sources also play in the innovation process; which is to say, other patents are by no means exhaustive of relevant knowledge required for new inventions. For example, even in the writing of this paper, multiple other concurrent papers appeared that also utilized text analysis methods which were made widely available through internet-based learning tools and resources such as Coursera and Stack Overflow.

One possibility is that inventors rely extensively on external knowledge in their ideas, which citations-based studies cannot capture, but is reflected in patent text. As more knowledge becomes easily and commonly accessible through the internet, we may see that the use of external knowledge "homogenizes" innovation across locations. [41] even suggested that geographic location would matter less through "cheapening. . . means of communication." This effect may be significant even if inventor networks are local or microgeographic clusters exist. In (Table 8), I find that similarity for patents that share non-patent citations (available from PatentsView) are much higher, which supports the claim that patent text reflects the knowledge flows from external sources.

Another example is introduction of new terms into the patent corpus. As new technology are developed, references make their way into patents. Patents that are the first to contain the new term are assumed to share some external sources of knowledge about the new technology. S5 Table show that patents introducing new terms rarely cite any backward citations in common, but do exhibit some similarity in their text. For example, "Adenovirus" is a term for a

**Table 8. Average text similarity of patents that share at least one common non-patent citation (*I*(*Common NPC protect* ≥ 1) = *T*), compared with patents that do not (*I*(*Common NPC protect* ≥ 1) = *F*).** Because random patent pairs have very sparse citation relationships, this uses a sample of patents that already share at least one common patent backward citation, which is why the baseline comparison group already has a high level of similarity.

| | Average Similarity | | | |
|---|---|---|---|---|
| | **1975-85** | **1985-95** | **1995-05** | **2005-15** |
| $I(CommonNPC \geq 1) = F$ | 0.293 | 0.280 | 0.258 | 0.250 |
| $I(CommonNPC \geq 1) = T$ | 0.426 | 0.401 | 0.382 | 0.548 |
| *p*-value | 0.091 | 0.000 | 0.000 | 0.000 |
| *t*-value | 1.692 | 5.217 | 12.092 | 189.720 |

virus that causes many common infections, particularly respiratory illness. With the development of gene therapy technology in the early 1990s, the first patent applications containing the term adenovirus appeared in 1993. Gene therapy delivers "correct" genes inside affected cells, and adenoviruses are often used as carriers for the corrected genes. In 1993, thirteen adenovirus patents were applied for that were later granted. (N.B. Failed applications are not accessible via the USPTO). While all adenovirus patents apparently utilised some common external knowledge sources, the average number of backward citations that was shared was 0.03, which represented an average of 0.0% of backward citations made.

However, if the influence of external knowledge is not desirable in all cases. It is up to the researcher to decide if incorporating these influences is appropriate for the research question. Patent text provides a new window into knowledge relationships, but one that may be too wide for certain applications.

## Conclusion

This paper focuses on knowledge dynamics of ideas embodied by patent text. I contribute methodologically to the literature on text analysis of patents by using an unsupervised machine learning approach in generating vector representations of patent text. I also offer an alternative lens to examining knowledge and find that different pattens of localization of knowledge spillovers are uncovered when examining patent citations vs idea proximity. These findings add nuance to our understanding of localized knowledge flows and highlight the possible importance of common and non-patent knowledge in generating innovation.

In further research, I address the question of whether Marshall-Arrow-Romer spillovers (the concentration of technologically similar firms within a city) facilitates greater innovation compared to Jacobs spillovers (the presence of technologically diverse firms within a city). Patent text may provide a unique contribution to assessing the diversity of innovative knowledge, which may otherwise be difficult to measure. Text similarity (or rather its reverse, text distance) may also be particularly suited in identifying "novel" ideas (similar to [30] and [24]). This would be a useful tool for questions assessing incremental vs radical innovation, as text similarity can easily identify inventions that are very close to prior inventions, and those that are distinct.

As shown in the JTH replication and discussed in [22], patent vectorizations provide a powerful alternative to USPC classification in assessing technological relatedness across patents. This could generate more accurate technological clusters or neighbourhoods of patents. Such an application would be beneficial in analyzing the effect of intellectual property rights on cumulative innnovation, as in [42], in providing more precise indicators of growth or decline within a technological cluster. Another application could be a different angle on a similar question related to agglomeration and microgeography, by identifying the extent to which similar patents cluster within a location, related to [5]. This methodology and much of the discussion could also be easily applied to scientific and academic texts, in the examination of collaboration [43], the effect of patents on science [44, 45]), and the evolution of scientific knowledge [12]. I believe the tools and methods discussed in this paper would be beneficial to any researcher of knowledge and innovation.

## Supporting information

**S1 Appendix. Text cleaning.**
(PDF)

**S2 Appendix. Document Vectors (Doc2Vec).**
(PDF)

**S3 Appendix. Latent Dirichlet Allocation.**
(PDF)

**S1 Fig. Example of document term matrix.**
(TIFF)

**S2 Fig. Illustration of document vectors.**
(TIFF)

**S3 Fig. Example of a patent converted into a distribution over topics.**
(TIFF)

**S1 Table. Selected topics as outputted by LDA.**
(PDF)

**S2 Table. Regression results for JTH replication under different control selection methods.**
(PDF)

**S3 Table. Regression results for JTH replication under different control selection methods.**
(PDF)

**S4 Table. Regression estimates for the localization of idea proximity of patent claims text.**
(PDF)

**S5 Table. Comparison of similarity and backward citation overlap for new patents using new terms.**
(PDF)

## Acknowledgments

Many thanks to Petra Moser, Luis Cabral, Robert Seamans, and Walker Hanlon for their kind support and feedback. Thanks to Lawrence White, Chris Conlon, Paul Scott, Michael Roach, Deepak Hegde, Pierre Azoulay, Anusha Nath, Todd Schoellman, Alexander Oettl, Pian Shu, Frank T. Rothaermel, Peter Thompson, Sonia Gilbukh, Paul Goldsmith-Pinkham, Nic Kozeniaukas, Roxana Mihet, Peifan Wu, Chase Coleman, Daniel Stackman for helpful comments.

## Author Contributions

**Conceptualization:** Sijie Feng.

**Data curation:** Sijie Feng.

**Formal analysis:** Sijie Feng.

**Investigation:** Sijie Feng.

**Methodology:** Sijie Feng.

**Project administration:** Sijie Feng.

**Resources:** Sijie Feng.

**Validation:** Sijie Feng.

**Visualization:** Sijie Feng.

**Writing – original draft:** Sijie Feng.

**Writing – review & editing:** Sijie Feng.

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
