## [Decision Letter · Decision Letter 0]

16 Mar 2020

PONE-D-20-03424

The proximity of ideas: an analysis of patent text using machine learning

PLOS ONE

Dear Dr Feng,

Thank you for submitting your manuscript to PLOS ONE. After careful consideration, we feel that it has merit but does not fully meet PLOS ONE’s publication criteria as it currently stands. Therefore, we invite you to submit a revised version of the manuscript that addresses the points raised during the review process.

Both reviewers are enthusiastic about your submission, but they suggest several useful directions in which you could go to sharpen your argument and increase the impact and significance of your findings.  I encourage you to give consideration to each of these suggestions.

In addition, I note a number of minor problems of grammar that need to be addressed.  And finally, you need to include a list of references (which was not part of the original submission).

We would appreciate receiving your revised manuscript by Apr 27 2020 11:59PM. To enhance the reproducibility of your results, we recommend that if applicable you deposit your laboratory protocols in protocols.io, where a protocol can be assigned its own identifier (DOI) such that it can be cited independently in the future. For instructions see: http://journals.plos.org/plosone/s/submission-guidelines#loc-laboratory-protocols

We look forward to receiving your revised manuscript.

Kind regards,

Joshua L Rosenbloom

Academic Editor

PLOS ONE

Journal Requirements:

5. Please ensure that you refer to Figure 1-3 in your text as, if accepted, production will need this reference to link the reader to the figure.

6. Please include a copy of Table 9, 10, 11 which you refer to in your text on page 11 and 18.

Reviewers' comments:

Reviewer's Responses to Questions

**Comments to the Author**

1. Is the manuscript technically sound, and do the data support the conclusions?

Reviewer #1: Yes

Reviewer #2: Partly

2. Has the statistical analysis been performed appropriately and rigorously? 

Reviewer #1: Yes

Reviewer #2: Yes

3. Have the authors made all data underlying the findings in their manuscript fully available?

Reviewer #1: Yes

Reviewer #2: Yes

4. Is the manuscript presented in an intelligible fashion and written in standard English?

Reviewer #1: Yes

Reviewer #2: Yes

5. Review Comments to the Author

Reviewer #1: This work applies recent advances in NLP and ML (doc2vec in particular) that enable more accurate assessments of document similarity, to contribute to the discussion (and remaining bit of controversy) around whether patent citations can be used as a measure of knowledge flow. The author finds that knowledge flows (as measured by citations) remain after taking more accurate similarity into account. However, the author goes on to demonstrate (in a new twist in this literature, to my understanding) that much of this effect is due to the same patent lawyers (assumedly citing prior art they are familiar with). In summary, the author argues that patent citations can only be used as knowledge flows in limited research contexts, and that their magnitude is much less than previously argued.

I agree with the author that it would be good to use more than the abstract of the patents. Patent abstracts are notoriously unhelpful and obtuse and it would be very interesting to see what happens when the claims and descriptions are taken into account. I can’t come up with any arguments that abstracts only would be biased, but at least a limited sample of whole document analyses would be reassuring.

You would develop a stronger punchline if you could develop a simple quantification of your claim that citations overstate localization in spillovers. I personally would not mind if you made some assumptions and stated a rough number (of course, please make assumptions explicit). But a simple percentage would get your work more attention (e.g., I find that modeling patent similar reduces the JTH estimate by x%, furthermore, when similar patent lawyers are taken into account, it reduces the JTH effect by y%).

Can you really not estimate the effect of same lawyers vs. same locale? You have a lot of data. Many law firms operate in multiple markets. This would be helpful.

Picky point (feel free to ignore) – it would be nice to get this done in CPC classes.

Can you kill all evidence of local spillovers? That would be a high impact paper.

You do an excellent job of summarizing the literature. I suspect this will become a standard reference for doctoral students and entry to the literature.

Reviewer #2: I really like this paper. I just have one issue. In the abstract there is the following statement: "For patents in the same technology field, their text similarity is 0.02-0.06 times higher if they are located within the same city, compared to patents from other cities. This effect is much smaller than when knowledge transfers are measured using patent citations: local patents receive about 0.9-1.4 times more local citations than compared to non-local control patents"

This is a potentially interesting result, but I think as stated it goes a bit too far, given the analysis. The issue is that it's hard to know how comparable these numbers are, since document similarity is measured quite differently from citation. For one, document similarity is a continuous variable ranging from 0 to 1, while citation is a binary variable. Maybe document similarity simply has a lot less variance than citations and that is what accounts for the smaller localization effect? Can you standardize these measures somehow to make them comparable?

For example, maybe you could convert the similarity measure into a binary one calibrated to match the baseline probability two patents cite each other? Then you have citation-links and similarity links, and the probability two patents in the same MSA share a similarity link is directly comparable to the probability they have a citation link.

Or maybe there is a way to measure both the citations and document similarity measures in terms of standard deviations?

Alternatively, some discussion of why it's appropriate to directly compare citation probability with document similarity might suffice.

6. PLOS authors have the option to publish the peer review history of their article (what does this mean?). If published, this will include your full peer review and any attached files.

Reviewer #1: No

Reviewer #2: No

---

## [Author Response · Author response to Decision Letter 0]

15 May 2020

Response to Reviewers

The proximity of ideas: an analysis of patent text using machine learning

Dear Dr. Rosenbloom,

Many thanks in the quick and diligent review you and the two reviewers have provided. They were thought provoking suggestions. I have included my response to each reviewer’s comments below. I apologize for the lateness of the edits, COVID19 caused some disruptions that had to be taken care of first. 

R1

Comment 1: “I agree with the author that it would be good to use more than the abstract of the patents. Patent abstracts are notoriously unhelpful and obtuse and it would be very interesting to see what happens when the claims and descriptions are taken into account. I can’t come up with any arguments that abstracts only would be biased, but at least a limited sample of whole document analyses would be reassuring.”

I agree with this assessment and have included a brief section titled “Patent abstract vs claims text,” where I repeat the exact same regression exercises on the similarity of claims text. I find that they yield slightly higher effects of geographic localization: patent claims text from the same technology field and location are 0.02-0.05 S.D.s higher than patents from different locations, but is comparable to abstract text.The output from the regressions are reported in S4 Table.

Comment 2: Can you really not estimate the effect of same lawyers vs. same locale? You have a lot of data. Many law firms operate in multiple markets. This would be helpful. 

I believe this is already captured (albeit disparately) in a few exercises. The JTH replication selects a control from the same lawyer and different cities. This is shown to reduce the localization effect by approximately 60%, showing that lawyers may account for a substantial portion of localization estimates in citations. I also measure the effect of having the same lawyer on patent text similarity in Table 6. Having the same lawyer has a substantially larger effect on patent text similarity than location match. I include both lawyer match and lawyer dummies as fixed effects in the main results of Table 4.

Comment 3: Can you kill all evidence of local spillovers? That would be a high impact paper.

The effect of location match is insignificant in Table 7 for the 1980-85 cohort; however, since it remains significant for all other exercises and years, I don’t belabour the point.

R2

Comment 1: This is a potentially interesting result, but I think as stated it goes a bit too far, given the analysis. The issue is that it's hard to know how comparable these numbers are, since document similarity is measured quite differently from citation. For one, document similarity is a continuous variable ranging from 0 to 1, while citation is a binary variable. Maybe document similarity simply has a lot less variance than citations and that is what accounts for the smaller localization effect? Can you standardize these measures somehow to make them comparable? … Or maybe there is a way to measure both the citations and document similarity measures in terms of standard deviations?

Agreed. Most of the main regression results, in tables 4-7, are reported on normalized similarity. Coefficients are measured in terms of standard deviations. For the citations replication exercise in Table 2, I keep results in percentage form so it is comparable with the original results. I report the normalized results of this exercise in S2 Table and S3 Table, so that the coefficients represent the increase in the (standardized) percentage of local citations when the patent is also local. I also alter the results mentioned in the abstract, introduction, and other parts of the article to refer only to normalized results.

Comment 2: Alternatively, some discussion of why it's appropriate to directly compare citation probability with document similarity might suffice.

I discuss this issue in the section “Relationship between text similarity and knowledge”. I weigh the pros and cons of interpretability in measuring knowledge spillovers using citations vs document similarity, and leave it up to the reader to decide which measure is more appropriate under her particular application. For example, if accurate indications of direct knowledge flows for individual patent pairs are required for the research question, then citations may still be the more appropriate measure. I refrain from directly claiming the similarity can replace citations, only offering some evidence that results may differ under slightly different definitions and measurement of knowledge spillovers.

I thank you all for your comments and efforts in reviewing my paper. I hope this response finds you well.

Best,

Sijie (Fiona) Feng.

---

## [Decision Letter · Decision Letter 1]

4 Jun 2020

The proximity of ideas: an analysis of patent text using machine learning

PONE-D-20-03424R1

Dear Dr. Feng,

We’re pleased to inform you that your manuscript has been judged scientifically suitable for publication and will be formally accepted for publication once it meets all outstanding technical requirements.

As you will see, Reviewer #1 wishes you had extended yourself further, but I do not think this constitutes a basis to ask you for further revision you do not choose to make.

Kind regards,

Joshua L Rosenbloom

Academic Editor

PLOS ONE

Additional Editor Comments (optional):

Reviewers' comments:

Reviewer's Responses to Questions

**Comments to the Author**

1. If the authors have adequately addressed your comments raised in a previous round of review and you feel that this manuscript is now acceptable for publication, you may indicate that here to bypass the “Comments to the Author” section, enter your conflict of interest statement in the “Confidential to Editor” section, and submit your "Accept" recommendation.

Reviewer #1: (No Response)

Reviewer #2: All comments have been addressed

2. Is the manuscript technically sound, and do the data support the conclusions?

Reviewer #1: Yes

Reviewer #2: (No Response)

3. Has the statistical analysis been performed appropriately and rigorously? 

Reviewer #1: Yes

Reviewer #2: (No Response)

4. Have the authors made all data underlying the findings in their manuscript fully available?

Reviewer #1: Yes

Reviewer #2: (No Response)

5. Is the manuscript presented in an intelligible fashion and written in standard English?

Reviewer #1: Yes

Reviewer #2: (No Response)

6. Review Comments to the Author

Reviewer #1: So I like your paper but I am disappointed you did not run additional analyses and simply cited a robustness check in very old work as evidence you need not run additional analysis. 

Reviewer #2: (No Response)

7. PLOS authors have the option to publish the peer review history of their article (what does this mean?). If published, this will include your full peer review and any attached files.

Reviewer #1: No

Reviewer #2: No

---

## [Editor Report · Acceptance letter]

18 Jun 2020

PONE-D-20-03424R1 

The proximity of ideas: an analysis of patent text using machine learning 

Dear Dr. Feng:

I'm pleased to inform you that your manuscript has been deemed suitable for publication in PLOS ONE. Congratulations! Your manuscript is now with our production department. 

Kind regards, 

on behalf of

Dr. Joshua L Rosenbloom 

Academic Editor

PLOS ONE